# Systematic Review of Hepatitis C Virus Prevalence in the WHO Western Pacific Region

**DOI:** 10.3390/v14071548

**Published:** 2022-07-15

**Authors:** Jenny Iversen, Handan Wand, Po-Lin Chan, Linh-Vi Le, Lisa Maher

**Affiliations:** 1Kirby Institute, Faculty of Medicine, UNSW Sydney, Sydney, NSW 2052, Australia; jiversen@kirby.unsw.edu.au (J.I.); hwand@kirby.unsw.edu.au (H.W.); 2World Health Organization Regional Office for the Western Pacific, Manila 1003, Philippines; chanpo@who.int (P.-L.C.); leli@who.int (L.-V.L.)

**Keywords:** hepatitis C virus, viral hepatitis, prevalence, systematic review, Western Pacific Region, elimination

## Abstract

Background: This review aimed to identify hepatitis C virus (HCV) prevalence estimates among the general population and six key populations (people who inject drugs, men who have sex with men, sex workers, prisoners/detainees, Indigenous people, and migrants) in the World Health Organization Western Pacific Region (WHO WPR). Methods: Original research articles published between 2016 and 2020 were identified from bibliographic databases. Publications were retrieved, replicas removed, and abstracts screened. Retained full texts were assessed and excluded if inclusion criteria were not met. Methodological quality was assessed using the Johanna Briggs Institute critical appraisal checklist for prevalence data. Data on HCV exposure and active infection were extracted and aggregated and forest plots generated for each population by country. Results: There were no HCV prevalence estimates in any population for more than half of WPR countries and territories. Among the 76 estimates, 97% presented prevalence of exposure and 33% prevalence of active infection. General population viraemic prevalence was 1% or less, except in Mongolia. Results confirm the endemic nature of HCV among people who inject drugs, with estimates of exposure ranging from 30% in Cambodia to 76% in Hong Kong. Conclusions: Countries require detailed knowledge of HCV prevalence in diverse populations to evaluate the impact of efforts to support WHO HCV elimination goals. Results provide baseline estimates from which to monitor and evaluate progress and by which to benchmark future elimination efforts.

## 1. Introduction

In 2015, the WHO estimated that worldwide ~71 million people were living with chronic hepatitis C virus (HCV) infection [1], a global prevalence of 1.0% (95% uncertainty interval (UI) 0.8–1.1) [2,3]. In 2019, there were an estimated 542,316 HCV-related deaths, with HCV accounting for 15·3 million (95% UI 13.3–17.5) global DALYs or 0.6% (0.5–0.7) of total global DALYs. Acute hepatitis, cirrhosis, and liver cancer contributed 1.7% (0.9–2.5), 79.5% (76.1–82.7), and 18.9% (15.9–22.2) to DALYs due to hepatitis C, respectively [4].

In 2016, 13.7 million people were estimated to be living with HCV in the WHO WPR (Western Pacific Region) [5] with mortality for HCV (24.1 deaths/100,000) the highest of any WHO region [2]. By 2019, the number of people living with HCV had increased to an estimated 23.5 million or 0.71% prevalence [6] with the region including 25% of the top 20 countries globally for HCV-related deaths [5].

The WHO’s 2016 Global Health Sector Strategy on Viral Hepatitis aims to eliminate viral hepatitis as a public health threat by 2030 through a 90% reduction in new infections and a 65% reduction in mortality [1]. HCV surveillance and prevalence data are crucial to monitoring elimination efforts, including estimating the population living with HCV, documenting the number in need of treatment, and assessing the need for targeted screening and testing in high-prevalence sub-populations.

The study aimed to systematically review and synthesize epidemiological data on HCV prevalence in the general population in 27 WHO WPR countries and 10 territories or areas, and in people who inject drugs, men who have sex with men, sex workers, prisoners/detainees, Indigenous people, and migrants.

## 2. Materials and Methods

Research articles were retrieved from PubMed/MEDLINE, Embase, and Western Pacific Region Index Medicus (WPRIM) databases in January 2021. The search strategies combined the concepts ‘hepatitis C virus’, ‘prevalence’, and ‘WHO WPR geographic areas’ using controlled and natural vocabulary (Appendix A Table A1). Searches were limited to records published from 1 January 2016 to 31 December 2020. No language restrictions were applied.

All retrieved publications were transferred to a bibliographic data management system (EndNote™X8.2) and spreadsheet (Microsoft Excel 360^®^). Replica publications were excluded. Country profiles were sent to WHO Country Focal Points with a request for additional publications. The study protocol was registered with PROSPERO (registration number CRD42020223181).

### 2.1. Eligibility Criteria and Selection

Publications were screened according to the following inclusion criteria: the study was conducted in one of the 37 WHO WPR countries or territories; reported serological or molecular HCV prevalence for a population of interest; included data on the number or frequency of individuals exposed to HCV at a specific time or time-period; data were collected in 2010 or later, and published between 1 January 2016 and 31 December 2020; and the study was conducted in humans. Populations of interest were: (a) the general population, defined as people living in a defined geographic area, excluding children aged < 18 years and high-risk groups; (b) non-remunerated blood donors: (c) pregnant women; (d) people who currently inject or previously injected illicit drugs; (e) men who have sex with men; (f) sex workers, defined as female, male, and transgender adults, over the age of 18, who receive money or goods in exchange for sexual services, either regularly or occasionally, and who may or may not self-identify as sex workers [7]; (g) prisoners/detainees, defined as people detained in closed settings; (h) Indigenous people, defined as distinct ethnic groups with a culture that is associated with a specific geographic region; and (i) migrants, defined as internal or foreign-born migrants living in a WHO WPR country or territory.

Exclusion criteria were that the study was a clinical trial, case–control, qualitative or intervention study, case report, case series, editorial, commentary, letter to editor, author reply, animal study, conference abstract, or review or modelling study that did not provide original HCV prevalence outcomes; had fewer than 15 participants or reported data from the general population, blood donors or pregnant women with a sample < 100; HCV status was self-reported; reported prevalence outcomes for a high-risk patient group (for example people living with HIV or liver disease); or was conducted only among children.

### 2.2. Selection Process

Identified publications were retrieved and abstracts screened by one reviewer (JI) with 10% checking by a second reviewer (LM). The main reason for exclusion was recorded. Full text publications were independently assessed for relevance by two reviewers (JI/LM) and excluded if they did not meet the inclusion criteria. Citations in systematic reviews, policy-related publications, and modelling studies were manually searched to identify additional publications. WHO Country Focal Points for viral hepatitis in the WPR were contacted and identified an additional 82 studies which were assessed for relevance.

### 2.3. Data Extraction

The following data elements were entered into the spreadsheet: author(s); year of publication; title; journal; country, region and/or city; study population; study design/sampling method; recruitment year/s; sample size; study response rate; HCV testing methods; numerator: number of HCV antibody positive or HCV RNA positive participants; denominator: number of participants tested for HCV antibody or RNA; and statistical methods used to determine prevalence.

Where multiple publications reported on the same study, only the most comprehensive was retained. Where prevalence estimates were reported for the same population using the same methods across multiple time points, only data for the most recent time point were extracted. Where general population studies included children and data were stratified by age group, prevalence estimates were recalculated with children excluded.

### 2.4. Assessment of Quality and Risk of Bias

The methodological quality of included studies was assessed using the Joanna Briggs Institute (JBI) critical appraisal checklist for prevalence data [8]. Each publication was assigned a grade using a score of ‘1’ for ‘poor’, ‘2’ for ‘good’, and ‘3’ for ‘excellent’ for each of the nine JBI critical appraisal items, for an aggregate quality score (range 9–27). Two reviewers (JI/LM) independently assessed each study and where there was disagreement, discussion achieved consensus. Publications with an aggregate quality score less than 50% of the maximum score (aggregate quality score < 13.5) were considered poor-quality. Additional analyses were conducted to assess the impact of inclusion of poor-quality publications.

### 2.5. Data Analysis

Data were aggregated and forest plots with 95% confidence intervals (CIs) were generated by country and each population of interest for HCV exposure (HCV antibody) and active infection (HCV RNA). Where 95% CIs for prevalence measures were not reported in publications, these were calculated using the Clopper–Pearson method [9]. Numerators were also calculated where these were not provided in the text. Where data were either missing or only provided in aggregate form, reviewers contacted authors by email with a request to provide missing or disaggregated data. All statistical analyses were performed with Stata v.14.2 (StataCorp, College Station, TX, USA).

## 3. Results

A total of 1659 articles were retrieved, including 443 replica records that were removed (Figure 1). Abstracts of the remaining 1216 publications were screened, with 1031 publications excluded on abstract review. Among 185 publications retained for full text review, 63 were retained. An additional study was identified for retention through a search of citations in systematic reviews. From 82 studies identified by Country Focal Points, 2 met the criteria for inclusion and were retained.

A total of 66 publications were retained, resulting in 76 HCV prevalence estimates among sub-populations of interest retrieved from 12 WHO WPR countries and territories (Table 1). No estimates were identified for 25 countries and territories. Among the 76 estimates, 97% (74) were HCV antibody prevalence and 33% (25) were HCV RNA prevalence (Table A2 and Table A3). With the exception of pregnant women (two studies, I2 = 11%), severe heterogeneity (I2 > 98%) was observed for all populations of interest. Most studies were conducted in sub-national geographic regions. The median aggregate quality score of the 76 included studies was 18 (range 10–24). Three studies were considered poor-quality due to an aggregate quality score < 13.5.

### Prevalence of HCV Antibody and RNA

HCV antibody prevalence estimates among the general population or one of the proxy populations were available for 11 countries and territories. The median aggregate quality score of the 31 studies was 19 (range 15–24). Estimates for HCV antibody prevalence among the general population were typically low at ≤1.0%, with the highest prevalence observed in Mongolia (13.6%, Figure 2). Eleven estimates of HCV RNA prevalence among the general population or proxy populations were identified; however, HCV RNA was not detected in two of these studies. Where HCV RNA was detected, the prevalence among the general population ranged from <0.5% in the Hong Kong SAR, Japan, and Malaysia to 11.0% in Mongolia (Figure 3).

Seventeen estimates for HCV antibody prevalence among people who inject drugs were obtained from seven countries and territories. The median aggregate quality score was 17.5 (range 15–22) and no studies were assessed as poor-quality. HCV antibody prevalence among people who inject drugs ranged from 30% in Cambodia to 76% in the Hong Kong SAR. Seven estimates for HCV RNA prevalence among people who inject drugs were identified from three countries, ranging from 24% in Australia to 51% in New Zealand.

Eleven estimates for HCV antibody prevalence among men who have sex with men were identified from two countries. The median aggregate quality score was 17 (range 13–21). Results did not change when the two studies assessed as poor-quality were excluded. HCV antibody prevalence was 0.6% in China and 26.1% in Viet Nam. Three estimates for HCV RNA prevalence among men who have sex with men were identified from two countries; the prevalence was 0.2% in China and 17.5% in Viet Nam.

Eight estimates for HCV antibody prevalence among sex workers were obtained from two countries. The median aggregate quality score was 18 (range 10–19), including one study assessed as poor-quality. Seven estimates were among female sex workers, where pooled HCV antibody prevalence was 15.4% in Viet Nam and 0.7% in China, noting that prevalence was slightly higher at 0.8% when the poor-quality study was excluded. Two estimates for HCV RNA prevalence among female sex workers were obtained from two countries, where HCV RNA prevalence was 0.04% in China and 8.7% in Viet Nam. One estimate was among male sex workers, where HCV antibody prevalence was 3.7% in Viet Nam (quality score 17). No estimates of HCV RNA prevalence among male sex workers were identified.

Only one estimate for HCV antibody prevalence among prisoners/detainees was identified, where the prevalence was 24% in Australia (quality score 17). No estimates for HCV RNA prevalence among prisoners/detainees were identified. Only one estimate for HCV antibody prevalence among Indigenous people was identified, where the prevalence was 7.0% in a population of Li ethnic minority in Baisha County, China (quality score 18). Similarly, only one estimate for HCV RNA prevalence was identified among Indigenous people, where the prevalence was 2.7% among Yi people in a Yi autonomous prefecture of southwestern China.

Five estimates for HCV antibody prevalence among migrants were identified from three countries. The median aggregate quality score of these five studies was 17 (range 17–19) and no studies were assessed as poor-quality. HCV antibody prevalence among migrants ranged from 0.4% in China to 1.2% in Australia. One study reported HCV RNA prevalence among migrants in China; however, HCV RNA prevalence was 0%.

## 4. Discussion

Globally, an estimated 1% of people (62.5 million) are living with hepatitis C infection. Viraemic prevalence of HCV in the WHO WPR is estimated at 1% (14 million people). In this review, viraemic prevalence of HCV infection among the general population was ≤1% in most WHO WPR countries where data were identified (Australia, China, Japan, Malaysia, Republic of Korea, Singapore, Viet Nam, and Hong Kong SAR). While HCV antibody prevalence was comparatively high in Papua New Guinea at 4.2%, only one small study among blood donors (n = 1206) was identified [10] and this estimate may not be representative of the general population. HCV antibody prevalence was also comparatively high at 3.9% in Cambodia where, similarly, only one small study (n = 868) was identified [11], However, hepatitis C viraemic prevalence in this setting was substantially lower at 1.3%. Mongolia was a notable exception, with this review confirming the endemic nature of HCV in this setting [12].

HCV infection is endemic among people who inject drugs, with an estimated 52% of people who inject drugs exposed to HCV globally [13] and 39% living with HCV [14]. In this review, estimates of exposure to HCV infection among people who inject drugs were comparatively high in the Hong Kong SAR (76%), New Zealand (73%), Viet Nam (69%), and China (67%). In the remaining three countries with available data, exposure to HCV among people who inject drugs was <50% (Australia, 49%; Singapore, 36%; Cambodia, 30%). Among the three countries with available estimates for HCV viraemic prevalence among people who inject drugs, approximately one-third had cleared the virus in New Zealand (73% exposed vs. 51% with active infection) and China (67% exposed vs. 41% with active infection), with half clearing the virus in Australia (49% exposed vs. 24% with active infection).

The global prevalence of HCV among men who have sex with men is estimated at 3.4%, with considerable geographic variation likely due to variation in the prevalence of both injection drug use and HIV among this population [15]. Estimates for the prevalence of HCV infection among men who have sex with men were identified from only two countries. In China, the prevalence of HCV infection among men who have sex with men (0.6% exposed) was comparable to the general population (0.7% exposed). In Viet Nam, the prevalence of HCV infection in men who have sex with men was substantially higher than in the general population (1% exposed), with one in four (26%) men who have sex with men estimated to be exposed to HCV. This was likely due to overlapping risk factors, with 19% of the Vietnamese sample reporting illicit drug use, 6% injection drug use, and 15% living with HIV [16].

Although sex workers may face elevated risk of HCV infection through either sex or drug use, there is limited information on the global prevalence of HCV in this group. This review identified prevalence estimates for female sex workers in only two countries (China and Viet Nam) and only one estimate among male sex workers (Viet Nam). In China, the prevalence among female sex workers (0.7% exposed) was comparable to the general population (0.7% exposed), with no estimate identified among male sex workers. In Viet Nam, the prevalence of exposure among female sex workers was substantially higher at 15.4% and 3.7% among male sex workers, likely due to overlapping risk factors with 8% of the female Vietnamese sample reporting injection drug use and 14% living with HIV [17].

Very few studies reported HCV prevalence estimates among prisoners/detainees (one study in Australia) or Indigenous people (two studies in China). In Australia, one in four prison entrants were exposed to HCV, likely due to overlapping risk factors [18]. In China, Li and Yi ethnic minorities appear to have a higher prevalence of HCV (7.0% exposed and 2.7% with active infection respectively) compared to the general population (0.7% exposed). Tattooing has been hypothesized as a potential route of transmission among Li people [19], while injection drug use is a probable route of transmission among the Yi ethnic minority [20]. Five studies reported HCV prevalence among migrants (Australia, the Republic of Korea, and three in China). In all three countries, HCV antibody prevalence among migrants was ≤1% and comparable to the general population.

There were significant gaps in the evidence, with no estimates of HCV prevalence among the general population, proxy populations, or key populations in 60% (16/27) of WHO WPR countries and 90% (9/10) of territories. There were no countries with available evidence of HCV prevalence among all populations of interest and only six countries and one member state had evidence for the prevalence of HCV among people who inject drugs, the group at highest risk of HCV infection and transmission. Gaps in the evidence for all populations of interest were particularly evident among the Pacific Islands and Territories.

While a strength of this review was the assessment of both exposure to HCV and viraemic prevalence, only one-third (25/76) of studies reported on viraemic prevalence. Among the estimates, there was significant heterogeneity for all populations of interest except for pregnant women, limiting comparability of results by country. Country-level estimates were pooled using source publication numerators and denominators, rather than prevalence. More weight was therefore given to studies with a larger sample size, further limiting the comparability of results between countries.

Countries and territories require robust, timely, and detailed knowledge of HCV prevalence and incidence in both the general population and key sub-populations to optimize and evaluate the impact of HCV prevention and treatment activities developed to support WHO HCV elimination goals. This review found limited studies with recent data on prevalence, with data only available for the general population, proxy populations, or key populations from 11 WHO WPR countries and 1 of 10 territories. HCV elimination efforts in the region, and especially in the 16 countries and 9 member states for which no estimates were identified, would be enhanced by implementing standardized seroprevalence surveys. Increased and improved data are particularly important for populations most at risk of infection and transmission where there are likely to be significant treatment benefits, such as people who inject drugs [21]. With the exception of one sub-population in one country (people who inject drugs in Australia), this review did not identify evidence of progress towards HCV elimination through hepatitis C treatment.

Systematic reviews of prevalence are becoming more conventional and have the potential to inform burden of disease policy and practice, including elimination efforts. However, their utility is limited by the availability and quality of existing data. Empirical estimates of HCV prevalence using direct methods are unavailable in many countries. Of available estimates, many are not recent, nationwide, or inclusive of key sub-populations. Along with improved efforts to develop systems for monitoring the epidemic and tracking progress towards elimination, the use of indirect methods is recommended to monitor prevalence and incidence and to validate elimination goals. Our results highlight the diversity of HCV infection among specific sub-populations in the WHO WPR, with a high prevalence of exposure to HCV among people who inject drugs and specific sub-populations where injection drug use is prevalent. Findings provide baseline prevalence estimates from which to monitor and evaluate progress and by which to benchmark ongoing elimination efforts, as well as a template for future reviews.

## Figures and Tables

**Figure 1 viruses-14-01548-f001:**
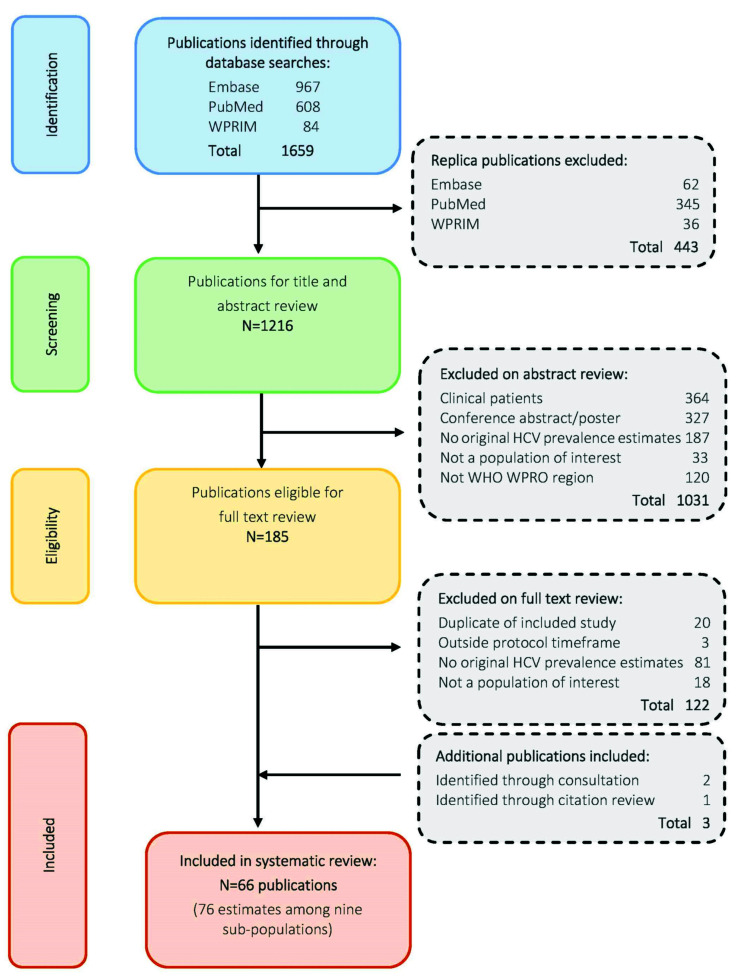
Flow chart of publication identification and selection process.

**Figure 2 viruses-14-01548-f002:**
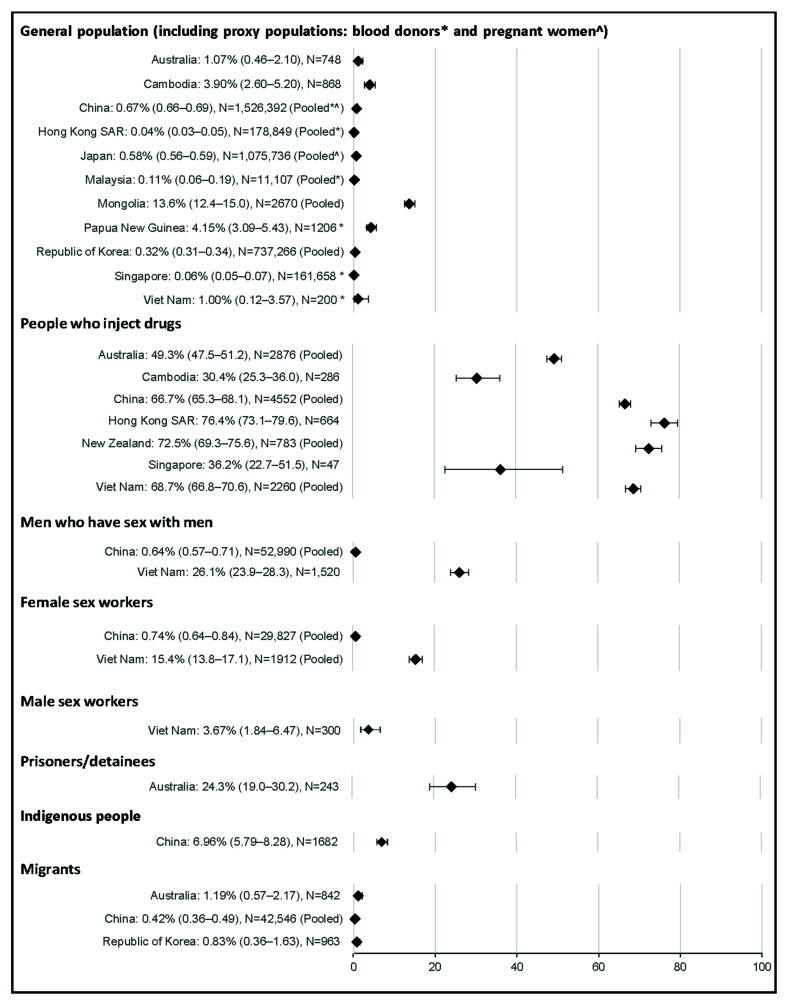
Forest plots HCV antibody prevalence by key population and country.

**Figure 3 viruses-14-01548-f003:**
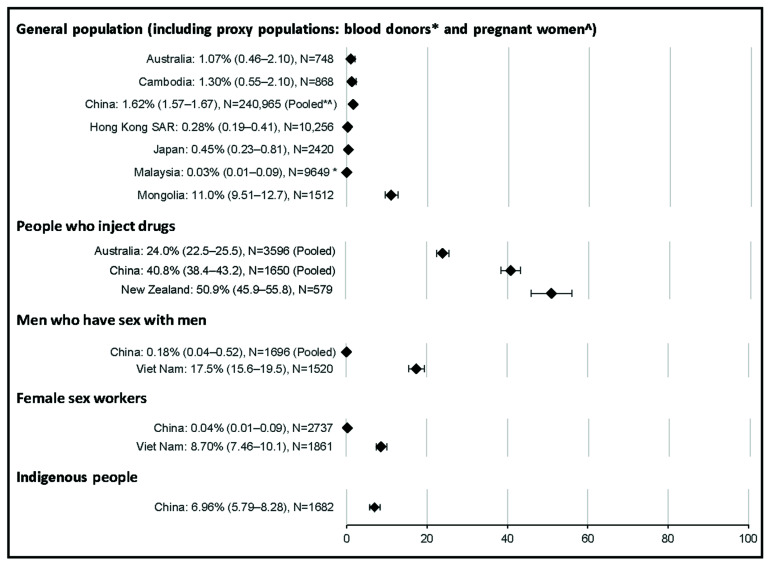
Forest plots HCV RNA prevalence by key population and country.

**Table 1 viruses-14-01548-t001:** Number of identified estimates for the prevalence of hepatitis C (anti-HCV or RNA) from WHO WPR by population category, 2016–2020 (n = 76).

Country	General Population	Pregnant Women	Blood Donors	People Who Inject Drugs	Men Who have Sex with Men	Sex Workers	Prisoners	Indigenous People	Migrants	Total
Australia	1			4			1		1	7
Cambodia	1			1						2
China	4	1	7	6	10	5		2	3	38
Hong Kong SAR	1		1	1						3
Japan	4	1							1	5
Malaysia	1		1							2
Mongolia	2									2
New Zealand				2						2
Papua New Guinea			1							1
Republic of Korea	3									4
Singapore			1	1						2
Viet Nam			1	3	1	3				8
**Total**	**17**	**2**	**12**	**18**	**11**	**8**	**1**	**2**	**5**	**76**

Note: No estimates for any population of interest were available for Brunei Darussalam, Cook Islands, Fiji, Kiribati, Lao People’s Democratic Republic, Marshall Islands, Federated States of Micronesia, Nauru, Niue, Palau, Philippines, Samoa, Solomon Islands, Tonga, Tuvalu, Vanuatu, American Samoa, Northern Mariana Islands, French Polynesia, Guam, Macao SAR, New Caledonia, Pitcairn Island, Tokelau, and Wallis and Futuna.

## Data Availability

Not applicable.

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
