# Peer review of "Systematic Review of Hepatitis C Virus Prevalence in the WHO Western Pacific Region"

_viruses, 2022, doi:10.3390/v14071548_

Round 1

Reviewer 1 Report

I read with interest the paper. It is well-written, methodologically it is very simple but robust. It's main limit is represented by the absence of a significant contribution in term of novelty. In fact, the aggregation of existing data has not revealed any substantial novelty with respect to the knowledge already available in this field.

Despite that, I think that overall the paper is interesting, and systematic reviews are always useful to highlight gaps in knowledge in a specific field. Thus I suggest to accept the paper.

I just have few suggestions/doubts:

- include in the eligibility criteria the upper time limit (31 december 2020, as reported in exclusion criteria);

- please revise the exclusion criteria avoiding to list those that are simply complementary to elegibility ones;

- I don't understand the sentence in section 3.1 "Eleven estimates of HCV-RNA prevalence among the general population or proxy population were identified; however, HCV-RNA was not detected in two of these studies". Thus, as I understand it, the execution of HCV-RNA is present in only 9 studies. Why do the authors say there are 11? What am I missing?

- Please revise the first part of the discussion. The fist page of discussion is just a summary/repetition of results. Discussion should not be a summary of results, but a discussion about it, with comments and hypothesis about the results themselves.

Reviewer 2 Report

Manuscript ID: viruses-1789791

Type of manuscript: Review

Title: Systematic Review of Hepatitis C Virus Prevalence in the WHO Western 

Pacific Region

Authors: Jenny Iversen, Handan Wand, Po-Lin Chan, Linh-Vi Le, Lisa Maher *

This was an informative systematic review. 

As authors mentioned, it was difficult to collect the updated HCV prevalence from many countries in the same period. In particular, special population; drug users, and pregnant women etc. 

Minor; 

I have one question for selection of reports. I could not identify the selected 66 papers. 

Did authors select the below paper? This paper showed the prevalence of HBV and HCV in 2015 and future trends. 

Tanaka J et al. Lancet Reg Health West Pac. 2022 Mar 16;22:100428. doi: 10.1016/j.lanwpc.2022.100428.
